# Predictive Factors for Bilateral Disease in Papillary Microcarcinoma: A Retrospective Cohort Study

Kirsten Lindner [1,†], K. Alexander Iwen [2,†], Jochen Kußmann [3] and Volker Fendrich [3,*]

1  Department of Endocrine Surgery, Lakumed, Krankenhausstr. 2, 84137 Vilsbiburg, Germany
2  Department of Internal Medicine I, University Hospital Schleswig-Holstein Campus Lübeck & Institute for Endocrinology and Diabetes–Molecular Endocrinology, Center of Brain Behavior and Metabolism CBBM, University of Lübeck, Ratzeburger Allee 160, 23562 Lübeck, Germany
3  Department of Endocrine Surgery, Schoen Clinic, Dehnhaide 120, 22081 Hamburg, Germany
*  Correspondence: vfendrich@schoen-klinik.de; Tel.: +49-40-2092-7101
†  These authors contributed equally to this work.

**Abstract:** Background: Based on risk stratification, the therapeutic options in papillary microcarcinoma (PTMC) can be active surveillance or surgery. Multifocal tumor occurrence can be decisive in determining the treatment strategy. The objective of this study was to identify risk factors for bilateral tumor occurrence in PTMC to enable individual therapy planning. Methods: A total of 545 PTMC patients who underwent thyroidectomy from 2008 to 2020 were retrieved. Univariate and multivariate analyses were performed to evaluate risk factors for bilateral PTMC. Results: 25.1% (n = 137) of all patients had multifocal PTMC, and 13.2% (n = 72) bilateral PTMC, respectively. In contrast to the maximum tumor size, the total tumor size significantly influenced a bilateral tumor manifestation (median total tumor size 5 mm versus 8.5 mm for bilateral PTMC, p < 0.001). A cut-off level for the total tumor size of >10 mm resulted in a sensitivity and specificity of 29.2% and 94.7%, respectively, in predicting a bilateral tumor manifestation, AUC 0.680 (95% CI, 0.611–0.748, p < 0.001). A cut-off of >4 tumors showed a sensitivity of 99.4% and a specificity of 97.5%, AUC 0.897 (95% CI, 0.870–0.924, p < 0.001) in predicting bilaterality. Conclusion: We could demonstrate for the first time that a total tumor size of >10 mm and more than four tumors significantly increased the risk of bilateral PTMC tumor involvement. These findings enable a risk-adjusted patient treatment.

**Keywords:** papillary microcarcinoma; bilaterality; multifocality; recurrence





## 1. Introduction

An increasing incidence of papillary thyroid microcarcinoma (PTMC), defined as papillary thyroid carcinoma measuring ≤1 cm, has been observed in the last decade. An occurrence of multifocal papillary microcarcinomas is described in 20–40% of patients with PTMC [1], and bilaterality in 16.2% of all patients [2]. The necessary therapeutic procedure has not yet been defined. The current ATA guidelines grade PTMC, whether solitary or multifocal, as a low-risk cancer group. Only multifocal PTMC with BRAF mutation and PTMC with extrathyroidal extension (ETE) are classified in the intermediary risk group [3]. Following this, according to the Consensus Statement from the Japan Association of Endocrine Surgery Task Force, multiplicity is not a contraindication for active surveillance [4]. In contrast, tumor size [5], extrathyroidal extension [6], tumor location [7], age [8], and multifocal tumor manifestation [1,9–11] are considered risk factors for a more aggressive tumor behavior, resulting in lymph node metastasis as well as tumor recurrence.

A patient evaluation of the National Thyroid Cancer Treatment Cooperative Study Group Registry, including 661 patients with intrathyroidal PTMC, revealed in a mean follow-up of 4.4 years that patients with multifocal PTMC with a reduced resection extent than a total or near-total thyroidectomy had a higher rate of recurrence than patients with

unifocal PTMC (18% versus 4%, $p < 0.01$) [1]. Xue et al. also supported a thyroidectomy in multifocal PTMC with an overall tumor size >10 mm, demonstrating improved relapse-free survival. In 88% of these patients (15/17), tumor recurrence occurred in the contralateral thyroid lobe [10]. However, whether the alleged tumor recurrence is not a metachronous but a synchronous tumor manifestation is difficult to determine.

Against this background, this retrospective study evaluates risk factors for a bilateral tumor manifestation in PTMC, focussing, in particular, on the number of tumor herds, the diameter of the largest PTMC, and the total diameter of all tumors.

## 2. Material and Methods

This retrospective single-center cohort study included patients who underwent thyroidectomy at the Department of Endocrine Surgery, Schoen Clinics, and the Department of Endocrine Surgery, LAKUMED, from January 2008 to December 2020, with the histological fuse of a PTMC (papillary thyroid carcinoma $\leq$1 cm). Finally, 545 patients were included (shown in Supplement Figure S1).

Analyzed data included demographic data, tumor size, TMN stage, number of tumors, and bilateral involvement. Based on the histologic findings, the number of tumors, the maximum tumor size, i.e., the diameter of the largest tumor, and the total tumor size, i.e., the sum of the diameters of all tumor foci, were evaluated.

Preoperatively, the extension of thyroid resection was planned according to the ultrasound findings. If suspicious nodules with a size >1 cm were shown sonographically, a fine-needle biopsy was added if necessary. This was not explicitly taken into account in the data collection of the present study. The indication for surgery was made on an individual basis, based on current guidelines, which were updated twice during the time period of the study.

A completion thyroidectomy was performed on 105 patients (19.3%) after receiving the histological diagnosis.

Chi-square was applied to identify significant differences. Differences in the mean of two samples were analyzed by an unpaired t-test. Multivariate analysis for bilateral tumor affection was calculated with a binary logistic regression model. Criteria for inclusion were significant on univariate analysis and clinical relevance (maximum tumor size). All confidence intervals (CI) reported are 95% confidence intervals. Furthermore, receiver operating characteristic (ROC) curves were generated to assess the diagnostic accuracy of maximum tumor size, total tumor size, and tumor number. The Youden index was used to identify cut-off values with the optimal sensitivity and specificity levels at which bilateral tumor affection can be distinguished from unilateral. $p$-values < 0.05 were considered statistically significant. Data were analyzed using IBM SPSS Statistic software for Mac (Version 26; IBM Corp., Armonk, NY, USA).

## 3. Results

### 3.1. Overall Collective

Our study included 545 patients with histological confirmed PTMC $\leq$ 1 cm (median age 51, range 13–83, women 97.1%) who underwent thyroidectomy from 2008 to 2020. A total of 25.1% ($n$ = 137) of all patients had a multifocal papillary microcarcinoma, 13.2% ($n$ = 72) had a bilateral location of a PTMC $\leq$ 1 cm.

Table 1 summarizes the demographic and histopathologic factors of the overall collective and in a subdivision of the presence of bilateral tumor detection. In comparing unilateral versus bilateral PTMC, bilateral involvement showed a significantly higher proportion of multifocal PTMC (100% versus 14%, $p < 0.001$). In addition, lymphatic metastases were more common (22.2% versus 11.4%, $p = 0.027$).

### 3.2. Bilateral Tumor Manifestation with One Solo Tumor in One Thyroid Lobe

To relate the risk based on the number of tumor foci in one thyroid lobe, only patients with bilateral tumor involvement and evidence of one or more tumor foci on one side and

a single tumor focus on the contralateral side were included in a subgroup analysis (*n* = 64) and compared to patients with unifocal PTMC (*n* = 473) (Supplement S1). The detection of two tumor foci in one thyroid lobe (*n* = 65) showed bilateral tumor involvement in *n* = 11 (16.9%), in three tumor lesions in 6 of 15 patients (40%), and four lesions in 5 of 8 patients (62.5%). Two patients suffered from five and six microcarcinoma in one lobe and had bilateral tumor evidence. Altogether, concerning the number of multifocal tumors and bilateral affection, an increased number of tumors had a significantly elevated risk of bilateral affection (*p* < 0.001). In addition, the total tumor size, defined as the sum of the diameters of all the tumor manifestations shown in one thyroid lobe, had a significant impact on bilaterality (5 mm versus 5.8 mm, *p* = 0.003). In contrast, maximum tumor size, defined as the size of the largest tumor manifestation, had no impact (5 versus 4 mm, *p* = 0.251) (shown in Figure 1).

**Table 1.** Patient characteristics.

| | Overall Collective (*n* = 545) | Unilateral PTMC (*n* = 473) | Bilateral PTMC (*n* = 72) | *p*-Value |
|---|---|---|---|---|
| Age | 51 (13–83) | 51 (13–80) | 53 (21–78) | 0.516 |
| Male | 114 (20.9%) | 92 (19.5%) | 22 (30.6%) | **0.026** |
| Lymphadenectomy (yes) | 139 (25.5%) | 119 (25.2%) | 20 (27.8%) | 0.365 |
| Completion TE (yes) | 105 (14.9%) | 78 (16.5%) | 25 (34.7%) | **<0.001** |
| Tumor localization | | | | |
| Right | 262 (48.1%) | 262 (55.4%) | 0 | |
| Left | 211 (38.7%) | 211 (44.6%) | 0 | **<0.001** |
| On both sides | 72 (13.2%) | 0 | 73 (100%) | |
| *n* stage | | | | |
| pN0 | 176 (32.3%) | 151 (31.9%) | 25 (34.7%) | |
| pN1 | 70 (12.8%) | 54 (11.4%) | 16 (22.2%) | **0.027 \*** |
| pNx | 299 (54.9%) | 268 (56.7%) | 31 (43.1%) | |
| Multifocal MPTC (yes) | 137 (25.1%) | 66 (14.0%) | 72 (100%) | **<0.001** |
| Number of tumors | | | | |
| 1 | 407 (74.7%) | 407 (86.0%) | 0 | |
| 2 | 96 (17.6%) | 54 (11.4%) | 42 (58.3%) | |
| 3 | 18 (3.3%) | 9 (1.9%) | 9 (12.5%) | **<0.001** |
| 4 | 15 (2.8%) | 3 (0.6%) | 12 (16.7%) | |
| 5 | 8 (1.5%) | 0 | 8 (11.1%) | |
| 7 | 1 (0.2%) | 0 | 1 (1.4%) | |
| Number of tumors (median, range) | 0 (0–7) | 0 (0–4) | 2 (2–7) | **<0.001** |
| Max tumor size (mm) | 5 (0.5–10) | 5 (0.5–10) | 4 (0.8–9.1) | 0.331 |
| Total tumor size (mm) | 5 (0.8–56) | 5 (0.8–56) | 7.1 (1–23) | **<0.001** |

MPTC = papillary microcarcinoma, TE = thyroidectomy, max = maximum, * pNx excluded, mm = millimeter.

Single-factor variables with *p* < 0.10, such as gender, lymph node metastasis, tumor count, total tumor size, and maximum tumor, were included in the logistic regression model. The results showed that the number of tumors and total tumor size were independent risk factors for bilateral tumor manifestation in patients with papillary thyroid microcarcinoma (shown in Table 2).

The maximum tumor size, total tumor size, and tumor number were evaluated using the ROC curves to identify a potential cut-off value as a risk factor for bilaterality. Here too, in contrast to total tumor size, maximum tumor size had no impact on bilaterality (AUC 0.680 versus 0.460, shown in Table 3). The number of tumors had the greatest impact (AUC 0.897) (shown in Figure 2).

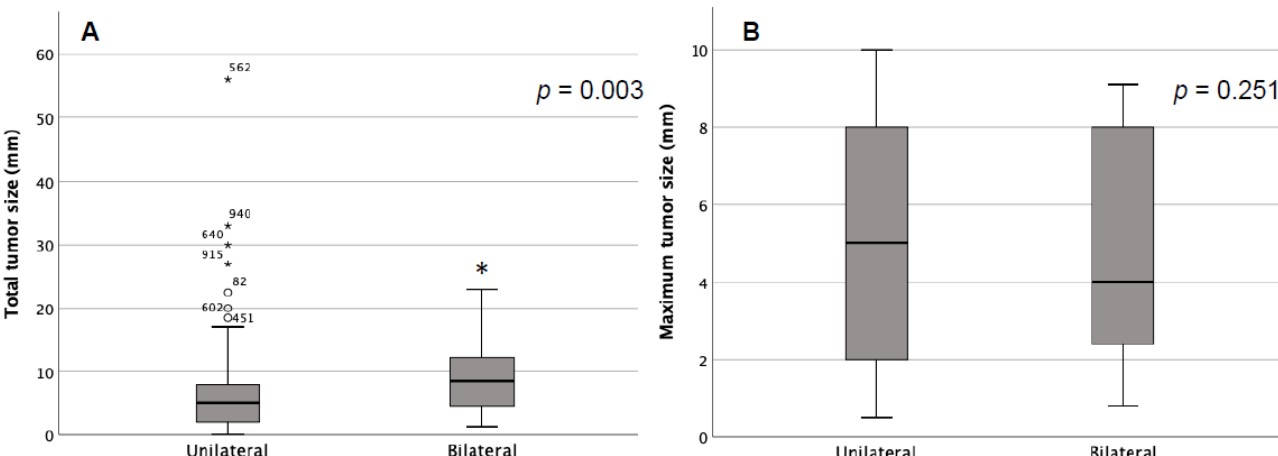

**Figure 1.** Total tumor size (**A**) and maximum tumor size (**B**) in unilateral and bilateral tumor manifestation. * statistically significant.

**Table 2.** Logistic regression analysis for potential risk factors of bilateral tumor spread in papillary microcarcinoma.

| Parameter | HR | 95%CI | | $p$ |
|---|---|---|---|---|
| | | Lower | Upper | |
| Sex | 0.443, 0.549 | 0.187, 0.316 | 1.052, 0.952 | 0.065, 0.074 |
| N stage | 1.000, 0.582 | 0.999, 0.888 | 1.001, 3.605 | 0.946, 0.103 |
| Tumor number | 2.282, 6.507 | 5.670, 4.354 | 19.922, 9.725 | **<0.001** |
| Tumor size max | 1.284, 0.958 | 1.000, 0.878 | 1.647, 1.046 | 0.050, 0.341 |
| Total tumor size | 0.772, 1.113 | 0.654, 1.060 | 0.910, 1.169 | **<0.001** |

HR hazard ratio, CI confidence interval.

**Table 3.** ROC Analysis of the predictive value of the risk factors for total tumor size, Maximum tumor size, and Tumor number for bilaterality.

| | AUC | SE | $p$-Value | AUC 95% CI | |
|---|---|---|---|---|---|
| | | | | Lower Limit | Upper Limit |
| Total tumor size | 0.680 | 0.035 | **<0.001** | 0.611 | 0.748 |
| Max tumor size | 0.460 | 0.036 | 0.296 | 0.388 | 0.531 |
| Tumor number | 0.897 | 0.014 | **<0.001** | 0.870 | 0.924 |

AUC = area under the curve, SE = standard error, CI = confidence interval.

Youden index was used to identify cut-off values in predicting bilateral tumor manifestation. Concerning total tumor size, a cut-off value of >10 mm sensitivity and specificity were 29.2% and 94.7% for prediction of bilaterality. Concerning tumor numbers, a cut-off value of >4 resulted in a sensitivity of 99.4% and a specificity of 97.5% (shown in Table 4).

**Table 4.** Boundary values of different factors in predicting bilateral tumor manifestation in patients with papillary microcarcinoma.

| | Cut-Off | TPR | TNR | YI | PPV | NPV |
|---|---|---|---|---|---|---|
| Total tumor size (mm) | >10 | 0.292 | 0.947 | 0.239 | 0.292 | 0.964 |
| Max tumor size (mm) | >8 | 0.125 | 0.82 | 0.055 | 0.625 | 0.883 |
| Tumor number | >4 | 0.994 | 0.975 | 0.86 | 100 | 0.882 |

TPR, true positive rate (sensitivity); TNR, true negative rate (specificity); YI, Youden's index; PPV, positive predictive value; NPV, negative predictive value; mm, millimeter.

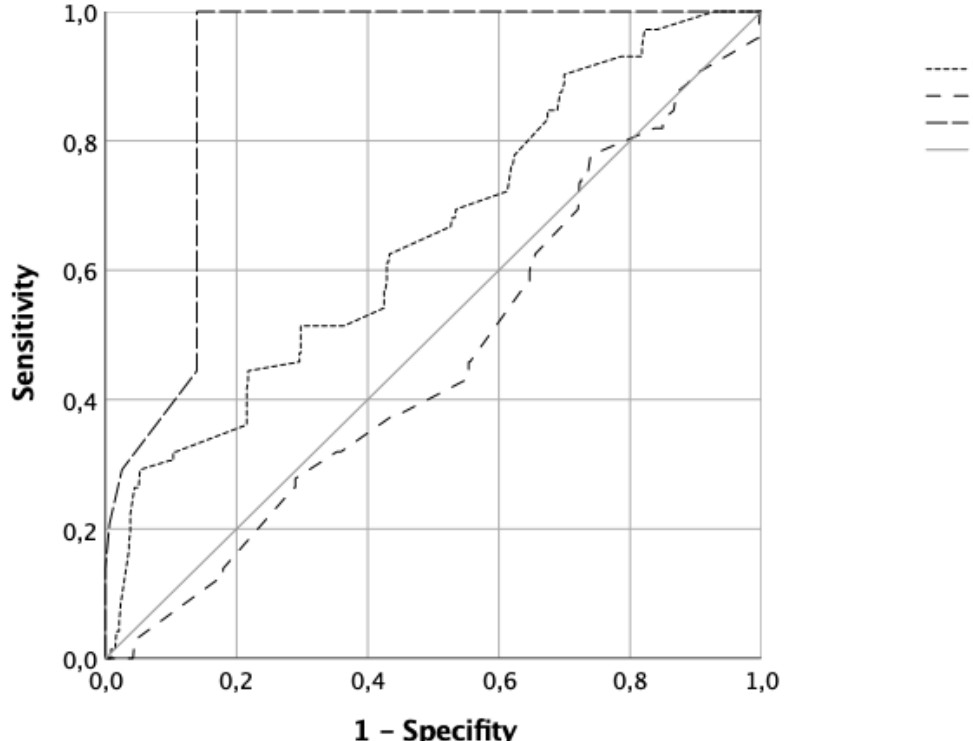

Diagonale Segmente ergeben sich aus Bindungen.

**Figure 2.** ROC curve analysis: sensitivity and specificity of tumor number, total tumor size, and maximum tumor size in predicting bilaterality.

## 4. Discussion

According to the ATA guidelines, a multifocal finding alone does not primarily increase tumor aggressivity in PTMC. However, the influence of multifocal tumor manifestation is discussed controversially in the current literature [12–14]. So which parameters can then be decisive—is the number of multifocal herds crucial or the tumor size? Our study showed that both the number of tumors and the total tumor size significantly influence bilateral tumor involvement of papillary microcarcinoma. The number of more than four tumor foci and a total tumor size of more than 10 mm were determined as clear cut-off values.

The risk of bilateral tumor affection was significantly connected to the tumor number. All patients with ≥5 tumors in one lobe had a tumor involvement on the contralateral side, in the case of four tumors, 62.5%, and 35.3% in three tumors. This goes in line with a study by So et al. The authors analyzed 277 PTMC patients and found multifocality in 36%. A contralateral tumor occurred in 30% of all patients with two tumors and in 46% with three tumors [15]. A study by Qu et al. focusing on multifocal PTC demonstrated that tumor behavior became increasingly aggressive with an increasing number of tumors [16]. Compared to the present study, the authors did not differentiate between tumor count in each thyroid lobe but referred to the total number of tumor foci on both sides. For the decision of a hemithyroidectomy versus thyroidectomy, this has little meaning. In our study, ROC analysis evaluated a cutoff value of more than four tumors as a significant predictive factor for bilateral tumor affection.

Lee et al. revealed a maximum tumor size of ≥1 cm as a predictive factor for contralateral carcinomas. Including 466 patients, bilaterality was demonstrated in 36.8% of all PTC patients (*n* = 174) and 25.7% of all PTMC patients (*n* = 292). Focusing on PTMC, both multifocality and tumor size (≤5 mm versus >5 mm) significantly impacted bilaterality [17]. In a study focusing on PTMC, Kaliszewski et al. identified a maximum tumor size ≥5 mm as a risk factor for bilaterality with a reported sensitivity of 69% and a specificity of 100%. A limiting factor was the small sample size of 177 patients, including 15 patients with bilat-

eral carcinoma [18]. A cutoff value of ≥5 mm was also postulated by Karatzas et al. [19]. Zhou et al. revealed a tumor size of ≥7 mm independently associated with bilateral PTMC. Here, too, the study collective is rather manageable, with 211 patients, including 54 patients with bilateral tumor involvement [20]. In our study, the maximum tumor size had no impact on bilateral tumor affection—in contrast to the total tumor size. To our knowledge, no study so far has focused on this aspect. However, in the same direction, a large study by Liu et al., including 1312 patients, demonstrated the significant impact of total tumor diameter compared to the unifocal tumor size on survival. They observed a reduced recurrence-free survival in PTMC with multifocal total tumor size >1 cm versus maximum tumor size ≤1 cm in a 10-year follow-up [21]. This goes in line with Feng et al. In their study, the authors demonstrated a greater aggressiveness in multifocal PTMC with a total tumor size of >1 cm with a greater portion of extrathyroidal extension and central lymph node metastasis. An exact number of tumor herds and a specification of tumor size are missing. Moreover, the low number of patients, including 19 patients with total tumor diameter ≤1 cm and 47 >1 cm, is a limiting factor [22].

Without question, active surveillance is an excellent therapy option for patients with PTMC. Even multifocal tumor involvement does not seem to negatively influence tumor progression under surveillance [23,24]. However, to date, only a few studies focused on this, including a study by Ito et al. on 1235 patients, including 147 (12%) with a multifocal tumor manifestation [25]. An appropriate patient selection remains one of the decisive factors for the success of the therapy. In this context, studies unquestionably demonstrated a more aggressive tumor biology in the case of multifocal PTMC [2,10,13,16,26]. In line with this, we demonstrated a significant impact of the summarized tumor size on the prediction of bilateral carcinoma. Our evaluated cut-off value of 10 mm in the ROC curve analysis aligns with the discussion of a higher classification of multifocal PTMC based on the total tumor size in T1b tumors [13,14]. While multifocal tumor involvement is often discussed in the literature, a more differentiated view is usually missing. However, a precise risk assessment of a bilateral tumor manifestation is essential to determine an individualized therapy strategy. On the one hand, this can be the question of the extent of the thyroid resection or, on the other hand, risk-adapted surveillance. Thus, the sum and number of suspicious foci should be clearly stated in the conclusions of the US report, as well as the presence of extracapsular spread, location, and suspicious lymph nodes (central/lateral compartments). However, the influence of multifocal PTMC on patient survival remains under discussion to date, and the influence of tumor multifocality on lymph node metastasis remains interesting.

Although this study is limited by its retrospective design, it is reasonably powered by including over 500 cases. The sonographic findings were not correlated with the pathological result, so no statements can be made on the preoperative detection of a PTMC. A study by Lacout et al. showed that when comparing the tumor size measured in the US versus the histopathological result in seventy-seven thyroid carcinomas, the median estimated US size was 7.52% too large—with a tumor size ≤10 mm ($n$ = 28), even 13% too large ($p$ = 0.054) [27]. Overall, additional studies would be helpful here. Due to the long study period, pathological processing was not carried out by a single expert, which can help to ensure comparability. Differentiated information on precise tumor location (isthmus, upper lobe), ETE, or molecular genetic processing, including BRAF analysis, is missing. Moreover, there is no follow-up for the patient. Thereby the impact of multifocal disease on patient outcomes remains unanswered. However, for the first time, this study focuses not only on the maximum or total tumor diameter but also on the number of tumor foci in papillary microcarcinoma in a thorough analysis.

## 5. Conclusions

The diagnosis of multifocal papillary microcarcinoma is not uncommon. The cut-off values evaluated in the present study for the total tumor diameter and the number of tumor

foci as predictive markers of bilaterality can be valuable complements for therapy planning in everyday clinical practice.

**Supplementary Materials:** The following supporting information can be downloaded at: https://www.mdpi.com/article/10.3390/curroncol29090473/s1, Figure S1: Overall collective.

**Author Contributions:** Study concept and design K.L., V.F., J.K. and K.A.I.; acquisition of data K.L. and K.A.I.; analysis and interpretation of data K.L. and K.A.I.; drafting manuscript K.L. and K.A.I.; critical revision of manuscript K.L., K.A.I., J.K. and V.F. All authors have read and agreed to the published version of the manuscript.

**Funding:** This research received no external funding.

**Institutional Review Board Statement:** The published data were collected for internal quality control; according to the local ethics committee, no specific approval was needed for publication. Please refer to the website of the Ethics Committee of the Hamburg Medical Association (https://www.aerztekammer-hamburg.org/sonstige_studien.html, accessed on 30 June 2022: "Ob eine Beratung durch die Ethik-Kommission der Ärztekammer Hamburg notwendig ist, hängt unter anderem von der Art der Datenerhebung ab. Bei der Erhebung pseudonymisierter Daten ist eine Beratung immer erforderlich. Bei der Auswertung zuvor anonymisierter Daten/Biomaterialien ist dies in der Regel nicht erforderlich".

**Informed Consent Statement:** Informed consent was obtained from all subjects involved in the study.

**Data Availability Statement:** The data presented in this study are available on request from the corresponding author.

**Conflicts of Interest:** The authors have no relevant financial or non-financial interest to disclose.

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
