# Peer review of "Predictive Factors for Bilateral Disease in Papillary Microcarcinoma: A Retrospective Cohort Study"

_curroncol, doi:10.3390/curroncol29090473_

Round 1

Reviewer 1 Report

The study addresses an important question with a large sample size and clear statistics. It is also well structured and presented with the necessary data to clarify the results.

Also, the clear indication that total tumor size is predictive for bilaterality is a very important finding. 

Only in the discussion, the language style might be improved by a native speaker.

I recommend publication after minor language edition.

Reviewer 2 Report

Comments and suggestions to the authors

Lindner K et al studied 545 patients with PTMC who underwent thyroidectomy from 2008 to 2020 to identify risk factors for bilateral tumor occurrence in these patients. They found that, total tumor size of >10mm and more than 4 tumor foci are predictive for bilateral disease. The ms is well written, the design of the study is appropriate, and the cited references are relevant to the research topic. The study is retrospective, but the authors mentioned this limitation.  

Major remarks

-        The authors did not correlate bilateral disease with adverse clinicopathological features or the disease outcome. So, their data are insufficient to guide therapeutic options for patients with bilateral PTMC. Accordingly, they should modify their conclusion.

-        Lines 37-38: According to ATA 2015 guidelines, multifocal PTMC with (not or) extrathyroidal extension are classified in the intermediate risk group for recurrence or persistent disease. The sentence should be rephrased.

-        Table 2: How the “Total tumor size >10mm” could be a risk factor for bilateral disease with HR<1?

Minor remarks

-        Table 4: The abbreviations “TRP, true positive rate (sensitivity); TNP, true negative rate (specificity) is not correct. Also, are the values of PPV and NPV for the parameter “Total tumor size” correct? Since the sensitivity is very low the NPV should be also low and the PPV high…..

-        Figure 1: the lines are not distinct 

Reviewer 3 Report

This is a retrospective cohort study aimed at identifying risk factors for bilateral thyroid involevement in thyroid papillary microcarcinoma. 

The abstract is adequate, and has listed rationale and setting details, alongside the most mportant findings.  

The objectives of the study are presented clearly and the introduction section communicates the need for defining current treatment recommendations that are affected by identifying bilateral thyroid involvement. The STROBE guidelines are missing, and a checklist should be added to the manuscript.

The Materials and Methods section has clearly defined inclusion and exclusion criteria, and has listed adequate statistical tests.

The Results section communicated the findings well.

The discussion is well researched and illustrates the need for subgroup analysis in patients that require a more aggressive treatment plan.

I would congratulate the authors on a well-crafted study design and would endorse publication pending minor revision.
